# NeuralGrasps: Learning Implicit Representations for Grasps of Multiple Robotic Hands

**Ninad Khargonkar**[1]    **Neil Song**[2]    **Zesheng Xu**[1]    **Balakrishnan Prabhakaran**[1]    **Yu Xiang**[1]
[1]The University of Texas at Dallas    [2]St. Mark's School of Texas
{ninadarun.khargonkar,zesheng.xu,bprabhakaran,yu.xiang}@utdallas.edu
23songn@smtexas.org

**Abstract:** We introduce a neural implicit representation for grasps of objects from multiple robotic hands. Different grasps across multiple robotic hands are encoded into a shared latent space. Each latent vector is learned to decode to the 3D shape of an object and the 3D shape of a robotic hand in a grasping pose in terms of the signed distance functions of the two 3D shapes. In addition, the distance metric in the latent space is learned to preserve the similarity between grasps across different robotic hands, where the similarity of grasps is defined according to contact regions of the robotic hands. This property enables our method to transfer grasps between different grippers including a human hand, and grasp transfer has the potential to share grasping skills between robots and enable robots to learn grasping skills from humans. Furthermore, the encoded signed distance functions of objects and grasps in our implicit representation can be used for 6D object pose estimation with grasping contact optimization from partial point clouds, which enables robotic grasping in the real world [1].

**Keywords:** Robot Grasping, Neural Implicit Representations, Grasp Transfer, Grasping Contact Modeling, 6D Object Pose Estimation

## 1 Introduction

Robot manipulation is a fundamental research problem in robotics. If we want to have robots that can perform tasks to assist humans autonomously, we need to enable robots to grasp and manipulate objects and use tools to perform tasks. Robots have different grippers ranging from two-finger parallel grippers to five-finger anthropomorphic robotic hands. Usually, manipulation research typically focuses on one type of robot gripper. For instance, two-finger grippers are widely studied due to their simplicity in planning and control. Most commercial robots designed for research adopt two-finger grippers such as the Franka Emika Panda arm, the Fetch mobile manipulator, and the Baxter robot. As a result, manipulation skills learned from these robots are limited to two-finger grippers. It is unclear how to transfer or share these manipulation skills to robots with different grippers, especially for skills learned from imitating humans or reinforcement learning [1, 2]. Similarly, skills learned for multi-finger grippers are not transferable to two-finger grippers [3].

In this work, we study how to transfer grasps between different robot grippers. First, grasp transfer enables robots with different grippers to share grasping skills. Second, we consider the human hand to be a special robot gripper, so we can transfer human grasps to robot grippers. This will enable robots to learn grasping skills from human demonstrations. Learning from human demonstrations is very valuable in semantic grasping [4] or task-orientated grasping [5] where robots need to grasp objects according to the tasks. To achieve this goal, we introduce a novel implicit representation of multiple robotic hands learned using deep neural networks. Our representation learning is motivated by DeepSDF [6] and Grasping Fields [7] which learn continuous Signed Distance Functions (SDFs) for 3D shapes of objects and human hands. We learn continuous SDFs for objects and multiple robotic hands. Our novelty lies in learning a latent space of grasps from multiple robotic hands, where a latent vector in this space encodes a grasp from a specific robotic hand. Importantly, the distance metric in the latent space encodes the similarity between grasps across different grippers.

---

[1]Dataset and code are available at https://irvlutd.github.io/NeuralGrasps

6th Conference on Robot Learning (CoRL 2022), Auckland, New Zealand.

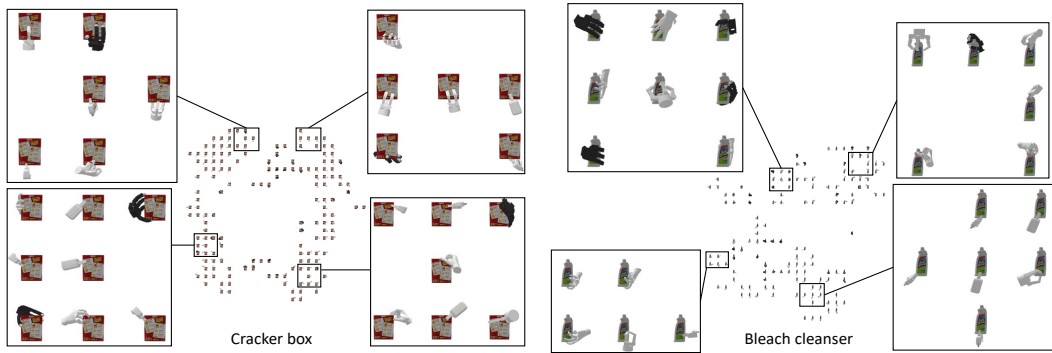

Figure 1: Barnes-Hut t-SNE visualization [9] using our learned embeddings of grasps from five different robotic hands.

Therefore, given a grasp from one gripper, we can retrieve the closest grasp from another gripper using the distance metric in the latent space. This property enables grasp transfer between different grippers including human hands. Fig. 1 shows the t-SNE visualization of grasps from five different robotic hands using our learned latent vectors for two objects. We demonstrate grasp transfer from human hands to a Fetch mobile manipulator in our experiments.

Specifically, we use a deep neural network trained to encode an object and a number of grasps from multiple grippers. We use an object-centric view where all the grasps are defined with respect to the object coordinate frame. Given a latent vector $\mathbf{z}$ and a 3D point $\mathbf{x} \in \mathcal{R}^3$ as input, the network predicts the SDF values of the 3D point to the object and to the grasp of a gripper corresponding to $\mathbf{z}$. During training, in addition to the loss function on predicted SDF values, we utilize a triplet loss function [8] to learn the distance metric in the latent space. In order to measure similarity between grasps from different grippers, we consider the contact maps of the grippers on the object, and use the similarity between contact maps to measure the grasp similarity. The triplet loss function extends the distance metric between contact maps to the latent space. In this way, the learned latent space preserves the similarity between grasps across different grippers.

Another advantage of our implicit representation is its use for real-world grasping. The learned SDFs of objects and grasps are fully differentiable functions. This property enables us to solve an optimization problem to estimate the 6D pose of object given a partial point cloud of the object as input. During this optimization, we also consider the contact of a grasp and the point cloud of the object. With the estimated 6D object pose, a robot can use the encoded grasps to grasp the object. We demonstrated that optimizing contact between a grasp and the input point cloud can improve grasping success rate in the real world. In summary, our contributions in this work are i) Presenting a novel implicit representation for grasps of multiple robotic hands; ii) Our representation enables grasp transfer between different robot grippers and human hands; iii) The representation can be used to solve object pose and optimize contact between grasps and observed point clouds in real-world grasping; iv) Novel grasping dataset with multiple robotic hands is introduced for multi-gripper transfer and to encourage future research.

## 2 Related Work

**Neural Implicit Representations.** Neural implicit representations for scenes and objects have attracted widespread attention recently. Occupancy networks [10] learn continuous occupancy functions of 3D shapes using neural networks. DeepSDF [6] trains neural networks to encode the Signed Distance Functions (SDFs) of 3D shapes, while Neural Radiance Fields (NeRFs) [11] encode both the geometry and color information of scenes into neural networks. The advantages of such implicit representations are: i) they are compact. For instance, a single DeepSDF network can encode a couple thousands of 3D shapes from the same category. NeRFs can encode large scenes with details with a single network. ii) The implicit representations are differentiable. So they can be easily used in gradient-based optimization to solve downstream tasks. For example, DeepSDF can be used for 3D reconstruction from partial observations. Grasping Fields [7] can synthesize humans grasps and reconstruct hands and objects jointly. One example of such implicit representations in robotics in-

cludes Neural Descriptor Fields [12] where implicit feature descriptors of 3D shapes are learned for robot manipulation, while in [13] the benefits of joint learning such neural representations with grasp affordances are shown. The advantages of such SDF-based object representations for object manipulation planning are shown in [14]. Such object representations have also been extended from rigid shapes to deformable objects, with ACID [15] using structured implicit representations for action-conditioned dynamics, and VIRDO [16] modeling the deformations on objects. Finally, NeRFs have also been utilized to obtain scene-level representations for training dense object descriptors [17] and models for dynamic 3D scenes [18]. The key difference in our work is the representation learning of grasps of multiple grippers and using it for both grasp transfer and object pose estimation.

**Grasping with Multiple Robotic Hands.** Traditional grasp planning methods such as the Graspit! simulator [19] use analytic approaches to access the grasp quality. These grasp quality measurements usually employ task wrench space analysis [20, 21] or force closure analysis [22, 23]. Analytic grasp planning methods can deal with different objects and different robotic hands. However, the main limitation of these traditional approaches is that they require full state information about objects such as shape and pose. They cannot work with partial observations of objects, e.g., point clouds from RGB-D cameras. Recent data-driven approaches for grasp planning utilize large-scale datasets with planned grasps [24, 25] and machine learning techniques to learn models that can work with partial observations [26, 27, 28, 29, 30]. However, majority of these works only focus on one type of robotic hand, especially, the two-finger parallel gripper. An exception is the UniGrasp [31] that considers gripper attributes in learning to detect grasping contact points on objects with a neural network. It can handle multiple grippers in detecting contact points for grasping. Our work differs from UniGrasp in that our implicit representations of objects and grasps enable grasp transfer between grippers and object pose estimation with grasp contact optimization. Another work closely related to ours is the ContactGrasp [32] that uses explicit contact maps to transfer grasps between human hands and robotic grippers. In contrast, our approach learns contact maps implicitly and uses a common latent space of grasps from human hands and robotic grippers for grasp transfer.

## 3 NeuralGrasps

Our goal is to learn representations over different robotic hands that can be used for downstream tasks such as grasping and grasp transfer. We leverage recent progress in neural implicit representations for representing 3D shapes. Specifically, DeepSDF [6] models the geometry of an object $o$ by learning the signed distance function $f_{\text{SDF}}(\mathbf{x}; o) : \mathcal{R}^3 \to \mathcal{R}$ over the object. Consequently, the signed distance function implicitly represents the surface points via the zero level set. DeepSDF models the 3D shapes of a set of objects by introducing a latent code for each object: $f_{\text{SDF}}(\mathbf{x}, \mathbf{z}_i)$, where $\mathbf{z}_i \in \mathcal{R}^d$ with $d$ the dimension of the latent space and $i = 1, 2, \ldots, N$ for $N$ shapes.

### 3.1 Grasp Encoding Network

Given a dataset of grasps from multiple grippers over an object $o$, we represent each grasp between a gripper and the object via the signed distance functions of both the gripper and the object in a normalized, object-centric 3D space. Motivated by DeepSDF [6] and Grasping Fields [7], we jointly model the two signed distance functions using a *grasp encoding* network. We utilize an auto-decoder based formulation as in DeepSDF where the network output is conditioned upon a latent vector $\mathbf{z} \in \mathcal{R}^d$ corresponding to each grasping scene in the dataset. So the grasp encoding network models the following function: $f_{\text{SDF}}(\mathbf{x}, \mathbf{z}; \theta_o) : \mathcal{R}^3 \times \mathcal{R}^d \to \mathcal{R}^2$, where $\theta_o$ denotes the network parameters corresponding to the model for object $o$. The output of the network is two SDF values for the query point $\mathbf{x}$: one for the object $f_{\text{SDF}}^o(\mathbf{x}; \mathbf{z}, \theta_o)$ and the other one for the gripper $f_{\text{SDF}}^g(\mathbf{x}; \mathbf{z}, \theta_o)$. The network architecture is illustrated in Fig. 2. Note that for a given grasp encoded by its latent code $\mathbf{z}$, the contact point set $\mathcal{C}$ between the gripper and the object is implicitly represented by the zero level set: $\mathcal{C} = \{\mathbf{x} \in \mathcal{R}^3 \mid f_{\text{SDF}}(\mathbf{x}, \mathbf{z}; \theta_o) = \mathbf{0}\}$. Because these are 3D points both on the surface of the gripper and on the surface of the object.

### 3.2 Grasp Similarity

We aim to constrain the latent vector $\mathbf{z}$ for a grasp to be close to the latent vectors of similar grasps. Since robot grippers have different kinematics, it is non-trivial to measure grasp similarity from different grippers. We consider an object-centric formulation here. We rely on the contact regions on the object to establish a similarity/distance measure over the grasp set. This follows the intuition that two *similar* grasps probably interact with similar regions of the object.

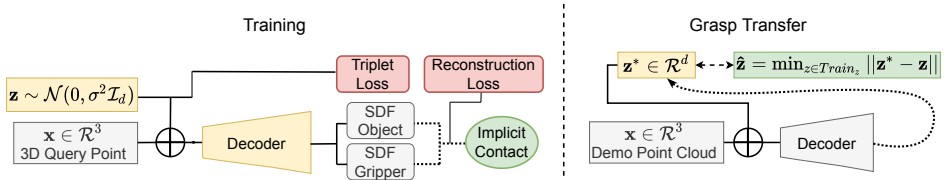

Figure 2: Left) Our network architecture with the training losses. Right) Illustration of the grasp transfer process during inference. Yellow modules indicate trainable parameters.

**Contact Map Representation.** To compare grasps on the basis of their contact regions on the object, we compute a contact map $\phi$ over the object points given a grasp. Given a grasping scene with two point clouds $P_o, P_g \subset \mathcal{R}^3$ representing the object points and the gripper points, respectively. Let $d(p_o, P_g)$ denote the distance between a point $p_o \in P_o$ to the closest point in $P_g$, i.e., $d(p_o, P_g) = \min_{p_g \in P_g} d(p_o, p_g)$, where we use the standard $L_2$ distance for $d(p_o, p_g)$ in our experiments. To obtain the object-centric contact map $\phi$, we simply take $\phi \in \mathcal{R}^{|O|}$ with each dimension corresponding to an object point $p_o \in P_o$. Then we define the contact map as $\phi(p_o; p_o \in P_o) = \exp(\frac{-d(p_o, P_g)}{\alpha})$, where $\alpha = 0.05m$ is a constant to penalize and score down object points with $d(\cdot, P_g)$ value higher than $\alpha$. Using this formulation, two grasps $g_1, g_2$ are considered to be similar if $\phi_{g_1} \sim \phi_{g_2}$. Note that the contact map is only assumed to be known at training time, and hence the unseen (test) grasps do not require a contact map for inference. A representative visualization of contact maps across different grippers and objects is shown in Fig. 3.

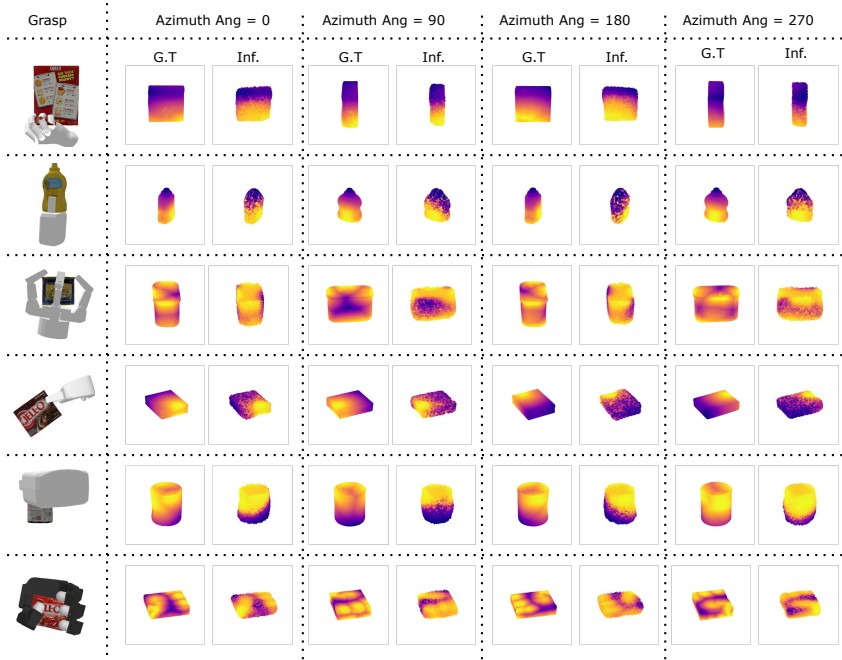

Figure 3: Multi-view visualization of ground truth (G.T) and inferred (Inf.) contact maps across different grippers and objects. Brighter regions correspond to the contact regions.

## 3.3 Learning the Implicit Representations

For a given object $o$, we train the network parameters $\theta_o$ on a dataset of grasps across multiple grippers. Each training sample $T_i = \{\phi_i, X_i\}$, $i = 1, \ldots N$, contains the contact map $\phi_i$ for a grasp and a set of (query, SDF) pairs: $X_i = \{(\mathbf{x}_i^j, SDF(\mathbf{x}_i^j))_{j=1}^M \mid \mathbf{x}_i^j \in \mathcal{R}^3; SDF(\mathbf{x}_i^j) \in \mathcal{R}^2\}$, where $M$ indicates the number of query points and $SDF(\mathbf{x}_i^j) = (SDF_o(\mathbf{x}_i^j), SDF_g(\mathbf{x}_i^j))$ consists of the ground truth signed distances to the object and the gripper, respectively. Similar to the auto-decoder formulation in DeepSDF [6], the training proceeds by pairing each grasp (training sample) with a

latent vector $\mathbf{z}_i$ and minimizing the loss function which consists of the two terms described below. We assume a zero-mean Gaussian as the prior over the latent vectors.

**Reconstruction Loss.** The reconstruction loss $\mathcal{L}_{\text{SDF}}$ is the $L_1$ distance between the ground truth and the predicted SDF values for the gripper and the object over an input query point. A standard practice in learning SDFs is to clamp both the ground truth and predicted distances to be within a specific range $[-\delta, \delta]$ with a clamp function: $\text{clamp}(x, \delta) := \min(\delta, \max(-\delta, x))$, where $\delta$ is a parameter used to control the learning of SDF to be within a certain distance of the surface ($\delta = 0.05m$ in our experiments). In this way, the learning focuses on the regions of the shape boundary. The reconstruction loss is defined as

$$\mathcal{L}_{\text{SDF}}(\mathbf{x}, \mathbf{z}, \theta_o) = |\text{clamp}(f_{\text{SDF}}(\mathbf{x}, \mathbf{z}; \theta_o), \delta) - \text{clamp}(SDF(\mathbf{x}), \delta)|. \tag{1}$$

**Triplet Loss in the Latent Space.** We constrain the latent vectors to represent a notion of similarity between the encoded grasps by utilizing the contact map $\phi$. Our goal is to encourage $\mathbf{z}_{g_1}, \mathbf{z}_{g_2}$ to be close to each other in the latent space if grasps $g_1, g_2$ are similar to each other on the basis of their contact maps $\phi_{g_1}, \phi_{g_2}$. We explicitly model such a constraint over the latent vectors during training since we use an encoder-less architecture. We make use of the triple loss [8] with a variable margin for each triplet of latent vectors $(\mathbf{z}_a, \mathbf{z}_p, \mathbf{z}_n)$. The triplet $(a, p, n)$ represents the anchor, the positive and the negative training samples. We use the similarity between contact maps to define positive and negative examples w.r.t an anchor example. The triplet loss is defined as

$$\mathcal{L}_{\text{triplet}}(\mathbf{z}_a, \mathbf{z}_p, \mathbf{z}_n) = \max\{d(\mathbf{z}_a, \mathbf{z}_p) - d(\mathbf{z}_a, \mathbf{z}_n) + m(a, p, n) , \ 0\}, \tag{2}$$

where $d(\cdot, \cdot)$ is the $L_2$ distance and $m(a, p, n)$ is the margin. In the original triplet loss function, the margin is a constant. In our case, we define the margin according to the similarity between grasp contact maps. Therefore, we can induce the grasp similarity into the latent space. The variable margin is defined as $m(a, p, n) = ||\phi_a - \phi_n||_2 - ||\phi_a - \phi_p||_2$ using $L_2$ distance between contact maps. Without loss of generality, given an anchor and two other grasps, we can select the one with smaller contact map distance as the positive example and the other one as negative example. Therefore, the margin $m(a, p, n)$ is always greater than 0 for every sampled triplet. For each training batch, we randomly sample a fixed number of triplets and compute the mean of the triplet losses.

**Training Optimization.** Overall, we solve the following optimization problem to estimate the network parameters $\theta_o$ and latent codes $\{\mathbf{z}_i\}_{i=1}^N$ for a set of $N$ grasps from multiple grippers:

$$\theta_o^*, \{\mathbf{z}_i^*\}_{i=1}^N = \arg \min_{\theta_o, \{\mathbf{z}_i\}_{i=1}^N} \Big[ \sum_{i=1}^N \sum_{j=1}^M \mathcal{L}_{\text{SDF}}(\mathbf{x}_i^j, \mathbf{z}_i, \theta_o) + \sum_{(a,p,n)} \mathcal{L}_{\text{triplet}}(\mathbf{z}_a, \mathbf{z}_p, \mathbf{z}_n) + \frac{1}{\sigma^2} \sum_{i=1}^N ||\mathbf{z}_i||_2^2 \Big], \tag{3}$$

Here, each grasping scene contains $M$ points and $\sigma$ is the standard deviation of a zero-mean multivariate-Gaussian prior distribution on the latent codes $\{\mathbf{z}_i\}_{i=1}^N$.

### 3.4 Solving Downstream Tasks with Our Implicit Representations

**Shape Reconstruction.** Given a latent vector $\mathbf{z}$ corresponding to a grasp, we can perform inference over the learned network to reconstruct the shape of the object and the shape of the gripper. This is achieved by querying a set of 3D points and then obtaining the object surface points $P_o = \{\mathbf{x} \in \mathcal{R}^3 \mid f_{\text{SDF}}^o(\mathbf{x}, \mathbf{z}; \theta_o) = 0\}$ and the gripper surface $P_g = \{\mathbf{x} \in \mathcal{R}^3 \mid f_{\text{SDF}}^g(\mathbf{x}, \mathbf{z}; \theta_o) = 0\}$. The contact points between the grasp and the object are intersection of the two sets of points. In cases when the latent vector $\mathbf{z}$ is unknown but we observe a set of points with their SDF values $(\mathbf{x}^j, SDF(\mathbf{x}^j))_{j=1}^M$, we can solve the following optimization problem to estimate the latent code:

$$\mathbf{z}^* = \arg \min_{\mathbf{z}} \Big[ \sum_{j=1}^M \mathcal{L}_{\text{SDF}}(\mathbf{x}^j, \mathbf{z}, \theta_o) + \frac{1}{\sigma^2} ||\mathbf{z}||_2^2 \Big]. \tag{4}$$

In the real world, the query points $\mathbf{x}^j$ are usually from depth sensors. We can approximate the SDF samples using a similar scheme as shown in [6] by sampling points at a small distance away from surface points along the normal. On obtaining the latent code $\mathbf{z}^*$, we can reconstruct the shapes of the object and the gripper.

**Grasp Transfer.** Due to the triplet loss imposed over the latent space, the inferred latent code $\mathbf{z}^*$ for an unseen grasp also follows the notion of grasp similarity. This metric learning constraint allows us

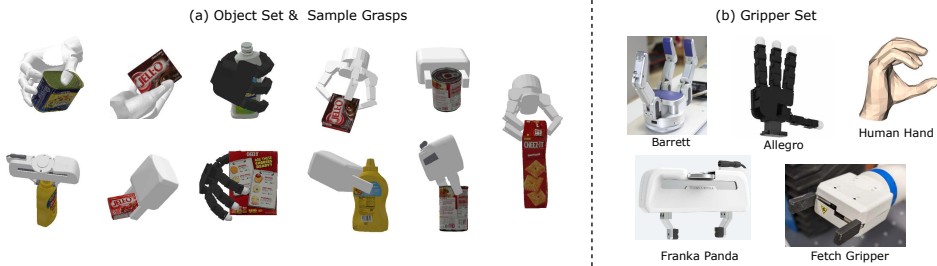

(a) Object Set & Sample Grasps

(b) Gripper Set

Barrett   Allegro   Human Hand

Franka Panda   Fetch Gripper

Figure 4: Objects and robotic hands in our multi-hand grasping dataset.

to retrieve a most similar grasp in the dataset on the basis of the nearest neighbor in the latent code space as shown in Fig. 2. The nearest neighbor grasp is not constrained to be from the input gripper, and hence such a framework makes it useful in a grasp transfer scenario where the demo and target grippers might differ. For example, the input grasp can be from a human as grasping demonstration.

**6D Object Pose Estimation with Grasp Contact.** Since our implicit representation encodes the SDFs of an object and its grasps, we can utilize it to estimate the 6D pose of the object, i.e., 3D rotation $R$ and translation $\mathbf{t}$, given camera observations in the real world. Suppose we can segment a set of 3D points of the object in the camera frame $\{\mathbf{x}_c^j\}_{j=1}^M$. If we transform these points into the object coordinate frame using $(R, \mathbf{t})$, they should have SDF values of the object close to zeros, since the transformed points are on the object surface. In addition, if the final goal is to execute a grasp $g$ to grasp the object, we can optimize the object pose such that the contact points of grasp $g$ are close to the observed 3D points. Let $\{\mathbf{x}_g^k\}_{k=1}^L \subset \{\mathbf{x}_c^j\}_{j=1}^M$ be the contact points of grasp $g$ in the camera frame. We solve the following optimization problem to estimate the object pose:

$$R^*, \mathbf{t}^* = \arg\min_{R,\mathbf{t}} \Big[ \sum_{j=1}^M |f_{\text{SDF}}^o(R\mathbf{x}_c^j + \mathbf{t}, \mathbf{z}; \theta_o)| + \sum_{k=1}^L |f_{\text{SDF}}^g(R\mathbf{x}_g^k + \mathbf{t}, \mathbf{z}; \theta_o)| \Big], \qquad (5)$$

where $f_{\text{SDF}}^o$ and $f_{\text{SDF}}^g$ are the SDF predictions of the object and the gripper from the network and $\mathbf{z}$ is the latent code for grasp $g$. Starting from an initial object pose, we can apply gradient descent to optimize object pose estimation. More details on object pose estimation are provided in the supplementary materials.

## 4  Multi-Hand Grasping Dataset

Most of the current datasets for object manipulation exclusively focus either on parallel jaw grippers [25] or human hand-object interaction [33]. Such datasets lack the data needed for learning neural implicit representations of multiple grippers. In order to train our proposed model, we generated a synthetic dataset of common robot grippers grasping objects. Creating a synthetic dataset of grasps over 3D models of real world objects allows for high detail grasping scenes containing the dense point clouds, contact regions, and signed distance function values.

**Grippers and Objects.** We selected five gripper models and seven objects from the YCB object set [34]. The gripper set includes four robot grippers across a range of the number of fingers: 2-finger Fetch and Franka-Panda grippers, 3-finger Barrett gripper, 4-finger Allegro finger, and finally a human hand model. The Fetch gripper was also used in our real-world experiments. The motivation to include a diverse set of multi-fingered grippers was to investigate how well such an encoding scheme works across grippers with different kinematic configurations. The seven objects from the YCB set are cracker box, tomato soup can, mustard bottle, pudding box, gelatin box, potted meat can, and bleach cleanser. Fig. 4 shows the objects and grippers grasps in our dataset.

**Grasp Generation.** To generate diverse multi-hand grasps over the YCB objects, we utilized the Graspit! simulator [19]. We initially generated a large number of grasps from GraspIt! and later sampled the initial set to generate a fixed amount of grasps for each gripper using the farthest point sampling algorithm which ensures diversity of gripper position with respect to the object. After the sampling stage, each grasping scene, i.e., an object with a gripper in a grasping pose, was loaded in the Pybullet [35] environment, and dense point clouds were rendered for the grasping scene using multiple viewpoints. Using the rendered point clouds, we generated the SDF values and the contact maps in the training samples as described in Section 3.3. We followed DeepSDF [6] to sample

Table 1: Chamfer distance ($\times$1e-3) for shape reconstruction and similarity scores for grasp retrieval

| YCB Model | Shape Reconstruction | | | | | | Grasp Retrieval | | | |
|---|---|---|---|---|---|---|---|---|---|---|
| | Object | Fetch | Barrett | Human | Allegro | Panda | Near Z | Near GT | Far Z | Far GT |
| *Cracker Box* | 1.33 | 1.44 | 5.52 | 2.86 | 3.03 | 2.48 | 0.81 | 0.88 | 0.19 | 0.15 |
| *Soup Can* | 2.60 | 4.95 | 4.06 | 3.21 | 3.97 | 4.20 | 0.73 | 0.84 | 0.13 | 0.09 |
| *Mustard Bottle* | 1.05 | 2.38 | 3.39 | 2.05 | 2.09 | 4.14 | 0.80 | 0.87 | 0.14 | 0.10 |
| *Pudding Box* | 1.49 | 5.27 | 8.27 | 5.45 | 3.80 | 5.40 | 0.77 | 0.84 | 0.25 | 0.16 |
| *Gelatin Box* | 1.20 | 5.42 | 7.04 | 4.72 | 3.74 | 3.72 | 0.75 | 0.81 | 0.22 | 0.14 |
| *Potted Meat Can* | 2.13 | 3.63 | 3.78 | 3.06 | 4.45 | 1.78 | 0.76 | 0.83 | 0.21 | 0.12 |
| *Bleach Cleanser* | 7.82 | 2.45 | 5.63 | 7.22 | 5.41 | 14.08 | 0.86 | 0.91 | 0.16 | 0.13 |

Figure 5: Grasp Retrieval: Query grasps from test set with close/far grasps from training set

majority of SDF values on query points close to the gripper and object surface. Our pipeline for data generation can be easily extended with additional sets of objects and grippers.

# 5 Experiments

## 5.1 Representation Learning on the Multi-Hand Grasping Dataset

To the best of our knowledge, there is no previous method that encodes grasps from multiple robotic grippers using neural implicit representation and enforces a notion of grasp similarity in a unified fashion. Hence, to validate our representation, we focus on the tasks of (1) shape reconstruction from a given latent code and, (2) grasp retrieval using the latent codes.

**Shape Reconstruction.** For this task, we consider a set of unseen test grasps with inferred latent codes from Eqn. (4). For each test grasp, query points are randomly sampled and passed as inputs to the network with the specific latent code. Using the predicted SDF values on the set of query points, the object and gripper points are separated out ($P_o, P_g$ in Section 3.4). We then compute the Chamfer distance against their ground truth point clouds and report the results in Table 1. The results shown in the table indicate a good performance even though reconstruction is not the primary goal of the method and is only included as a validation step for the representation learning. Furthermore, the inferred contact maps for such reconstructions align closely with the ground truth as seen in Fig. 3. The apparent difference between the two in terms of smoothness of the heatmap is due to the inferred contacts being shown on the reconstructed shape which approximates the original shape.

**Grasp Retrieval.** To test the effectiveness of the metric learning loss on the latent space, we perform an experiment on grasp retrieval for an input query grasp with Fig. 5 showing some results. We use the latent vector of an input query grasp and retrieve the closest and farthest grasps using (a) the distance between the latent vectors as a similarity measure, and (b) the distance between the contact maps which is considered as the ground truth (GT in Table 1). We report the similarity score $s = 1 - d$ ; $d \in [0, 1]$ between the query and retrieved grasps for both (a) and (b) in Table 1. $d$ is the normalized $L_1$ distance between the contact maps of the pairs of grasps. As seen in Table 1, the learned latent codes have similarity scores aligned with those for the contact maps based method, along both nearest and farthest metrics. This verifies that the learned grasp embeddings preserve the similarity metric of using contact maps.

## 5.2 Real-World Experiments on Grasping and Grasp Transfer

**Grasping with Object Pose Estimation.** To validate our implicit representation in the real world, we first conducted a grasping experiment with the 7 YCB objects in our dataset on a tabletop. A Fetch mobile manipulator is employed for grasping. For each trial, we simply put one object on a table and we use the table height to segment the point cloud of the object from the Fetch head camera. Using the segmented point cloud, we estimate the object pose according to Eqn. (5). Here, we first obtain an initial pose estimate using the object point

cloud and then we retrieve a grasp from the training set closest to the current gripper position for optimization. For comparison, we also tested a baseline model that only uses the SDF of the object without grasp contact optimization, i.e., no $f^g_{\text{SDF}}$ in Eqn. (5). Since Eqn. (5) requires a grasp for optimization, we first use the baseline model to estimate the object pose, and then select the closest grasp from the grasp set to the current gripper location for optimization.

Using the estimated pose, we attempt to grasp and lift the object after finding a suitable motion plan using MoveIt with 5 trials for each object. If the grasp execution resulted in the gripper not being to pick and hold the object, the trial was counted as a failure. The results of this study are shown in Table 2 where we can see the relative improvement in grasp success rate (overall fraction of successful trials) for the method using the contact-based pose estimation. This also shows the importance of the implicit contact points in the framework since contact is a critical component of grasping.

Table 2: Grasp success over 5 trials: baseline vs. grasp contact optimization

| YCB Model | Baseline | With Contact |
|---|---|---|
| *Cracker Box* | 5 | 5 |
| *Soup Can* | 2 | 4 |
| *Mustard* | 2 | 4 |
| *Pudding Box* | 4 | 4 |
| *Gelatin Box* | 3 | 5 |
| *Potted Meat* | 3 | 4 |
| *Bleach* | 3 | 4 |
| #Success | 22 | 30 |
| Success Rate | 0.628 | 0.857 |

**Human-to-Robot Grasp Transfer.** We consider the task of transferring human grasps to a Fetch robot via the proposed implicit representation. We first conduct object pose estimation as described above. Then a person demonstrates a grasp on the object. Using the RGB-D images from the Fetch camera, we estimate the 3D hand joints using A2J [36] and then utilize Pose2Mesh [37] to reconstruct the 3D hand mesh from the predicted 3D hand joints. We combine the hand points from the 3D hand mesh with the object point cloud and infer a latent code for the demonstrated grasp via Eq. (4). The inference optimization ran for 800 iterations and took about 7 seconds on average. Using the inferred latent code, we query for the closest Fetch gripper grasp in the encoding space of training data grasps and execute it in the real world. We show a qualitative result of such a grasp transfer from a human demonstration in Fig. 6. More examples can be found in the supplementary materials.

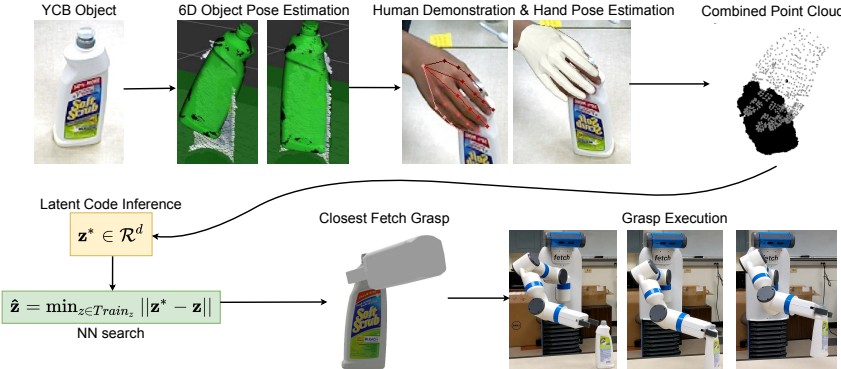

Figure 6: Our pipeline for grasp transfer from human demonstrations to a Fetch mobile manipulator.

## 6  Conclusion

We introduce NeuralGrasps, a framework for learning implicit representations for grasps of multiple robotic hands. In our framework, grasps across different robotic hands are embedded into a shared latent space, where the distance metric in the latent space is learned to preserve grasp similarity. Therefore, NeuralGrasps enables grasp transfer between grippers in the latent space. In addition, a latent code can be decoded to the signed distance functions of an object and a gripper in a grasping pose. This property enables us to perform 6D object pose estimation with grasping contact optimization using partial observations in the real world. With the estimated object pose, robot grasping can be achieved using the encoded grasps in NeuralGrasps. In the future, we plan to address several limitations in NeuralGrasps. First, we train a single network for each object in our method. We will study learning representations of multiple objects and multiple grasps with a unified network. Second, NeuralGrasps cannot synthesize novel grasps in the latent space well. We will extend it to grasp synthesis in the future. Lastly, we will train NeuralGrasps with a large number of objects and make it be able to deal with unseen objects during inference.

**Acknowledgments**

This work was supported in part by the DARPA Perceptually-enabled Task Guidance (PTG) Program under contract number HR00112220005. B. Prabhakaran's work was supported by (while serving at) the National Science Foundation.

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
