# OpenReview forum: "NeuralGrasps: Learning Implicit Representations for Grasps of Multiple Robotic Hands"
_robot-learning.org/CoRL/2022/Conference — CoRL 2022 Poster_

### Official Review · Reviewer_bHvy · 2022-07-17

**Originality:** Good
**Technical Quality:** Very Good
**Clarity Of Presentation:** Very Good
**Impact:** 3

**Recommendation:**

Weak Accept: I recommend accepting the paper, but will not argue for my recommendation if the majority of other reviewers have a different opinion.

**Summary:**

This paper proposes an algorithm to learn implicit neural grasp representations and presents some applications of these representations. The proposed algorithm builds on DeepSDF [6]. DeepSDF jointly optimizes model parameters and per-shape latent codes over a dataset of 3D shapes. A neural network function parameterized by the learnt model parameters predicts the SDF value from input 3D location and latent code.

Here this idea is extended to grasps by making the function output two SDF values - one for the object and one for the gripper in the grasping pose. Training data is automatically generated using GraspIt!. In addition to optimizing the per-grasp latent codes to minimize SDF prediction error, the latent codes are also encouraged with a triplet loss to cluster grasps with similar object contact patterns, regardless of the grasping gripper geometry.

Experiments evaluate cross-gripper grasp retrieval, object pose estimation for grasping, and grasp transfer from human to robot.

**Issues:**

Please address the "clarification" questions in the "weaknesses" section. Addressing the conceptual question by counter argument or by implementing and evaluating the the suggested two-stage learning process would be even better, but I don't think is necessary for acceptance.

**Quality Of The Limitations Section:**

Limitations are addressed clearly

**Reviewer Expertise:**

3: The reviewer is fairly confident that the evaluation is correct

**Robotics Focus:**

Sufficient demonstration on hardware

**Strengths And Weaknesses:**

**Strengths**
- The basic grasp representation idea presented here has many downstream possibilities because various grasp-level distance metrics can be imposed on this compact representation (the latent code). This paper demonstrates one possibility through an object contact map similarity metric. But one can imagine others e.g. temporal encoding for skills, gripper contact map similarity, etc.
- The paper is mostly well written and understandable. Figures are helpful.
- Real robot experiments show the practical application of the proposed algorithm.

**Weaknesses**

Conceptual: Inadequate motivation discussion for the proposed training procedure and choice of a single latent code in the paper.
- The SDF accuracy and cross-gripper contact map similarity objectives conflict with each other in the current formulation, because they share the same latent code. In the limit, training manages to drive the triplet loss to zero, that means two grasps with different grippers but similar object contact maps would have the same latent codes, and hence the same SDF predictions i.e. at least one gripper will be reconstructed incorrectly.
- This is seen in the significant inaccuracies of the inferred contact maps in Figure 3, which implies large gripper reconstruction inaccuracy. Table 1 chamfer distance numbers corroborate this.
- This raises the question, why the two objectives need to be 1) solved with the same latent code, and 2) optimized simultaneously. For example, how about training $\theta$ and latent codes firstly with just the SDF accuracy objective i.e. same as DeepSDF. Then a _separate_ MLP can be trained _separately_ to embed the DeepSDF latent codes into a different latent space with the proposed triplet loss. This would resolve the conflicting objectives mentioned above.

Experimental:
- Grasp transfer between various grippers is mentioned as a strong motivation (L29), but the real robot experiment shows only one simple top-down grasp.

Clarification:
- Section 5.2, "grasping with object pose estimation": Eq. 5 requires the latent code, it is not clear which latent code is used to optimize the object pose.
- Figure 3: Please show the contact maps from multiple viewpoints.
- Table 2: how the grasp success rate defined?

**Summary Of Recommendation:**

I am recommending to accept this paper because of the clear research question and good execution. The idea of imposing distance metrics on a neural grasp representation can be developed further for multiple applications.

---

> ### Author Response · Authors · 2022-08-23
> **Responses to Reviewer bHvy**
>
> We thank the reviewer for the insightful review and bringing about interesting points of discussion. We answer each of the required clarifications below.
>
> **Q1:** Conflict between the triplet loss and SDF loss terms while training the model.
>
> We agree that there is indeed a tradeoff between the two objectives w.r.t the triplet loss and SDF accuracy. However, we view the triplet loss on the latent codes as a form of regularization. As in variational auto-encoders, a normal distribution is usually employed to regularize the latent embeddings while minimizing the reconstruction loss. Hence, the two losses need to be optimized jointly to achieve an equilibrium.
>
> One of the reasons for why object reconstruction is better compared to gripper reconstruction is because the same object is being trained across multiple grippers, so naturally it has more training points compared to the different grippers.
>
> **Q2:** Motivation required for joint training and why not a two-stage separate training process to resolve the conflicting objectives.
>
> The reviewer’s suggestion to have a separate MLP to re-encode the latent vectors is certainly interesting. But in our view that comes at an increased complexity to the model and we lose the benefit of end-to-end training, since we then require a two-stage process. Finding the closest match will also then be a 2-stage process which is not exactly desirable. This is also motivated by previous metric learning works where the metric constraint is imposed jointly to reap the benefits of such joint training
>
> As the reviewer pointed out, the inferred and GT contact maps do not exactly match. But we would like to note that such result is expected with joint training and approximating the contact maps – hence the smooth color results for GT and some discontinuities for the inferred contact maps.
>
> **Q3:** The real robot experiment shows only one simple top-down grasp.
>
> Please see our supplementary video for more examples of the grasp transfer. We have conducted some experiments with different grasps beyond top-down grasping.
>
> **Q4:** It is not clear which latent code is used to optimize the object pose in Eq 5.
>
> Please see Section 3.2 in the supplementary material for more details about the 6D object pose estimation with grasping contact. We first perform an initial pose estimation of an object using the object point cloud. Then we find a Fetch gripper grasp from the training set closest to the current gripper position for optimization. The latent code is from this selected closest grasp.
>
> **Q5:** Figure 3: Please show the contact maps from multiple viewpoints.
>
> We have included the multiple viewpoint figures for the Figure 3 in the revised paper.
>
> **Q6:** Table 2: how the grasp success rate defined?
>
> The grasp success rate is defined as the number of times the gripper was able to successfully grasp the object and hold it. If the execution resulted in the gripper missing the object or not able to hold it, that was counted as a failure. Finally, to get the rate we divide the number of successful trials by total number of trials. We have also clarified this in the revised paper.

---

### Official Review · Reviewer_hW79 · 2022-07-28

**Originality:** Good
**Technical Quality:** Good
**Clarity Of Presentation:** Very Good
**Impact:** 3

**Recommendation:**

Weak Accept: I recommend accepting the paper, but will not argue for my recommendation if the majority of other reviewers have a different opinion.

**Summary:**

In this work, the authors proposed a new method that leverages 3D SDF-based implicit representation to learn an embedding for object grasps. The learned embedding can index various grasps of the same object but using different grippers, e.g. the two-finger parallel gripper, three or four-finger ones, and even the human hand, so that grasp transfers from one gripper to another or from human demonstration to robot execution are possible. Experiments are done using 7 YCB shapes and 5 different grippers. Simulated grasps are collected for training and real-world/real-robot experiments are presented.

**Issues:**

Please see the "weakness" points above. Major points to address are why learning a latent embedding is beneficial, can you solve for the grasp poses given a new gripper instead of doing retrieval, and can you make the system not overfitting to one shape.

**Quality Of The Limitations Section:**

Limitations are addressed clearly

**Reviewer Expertise:**

4: The reviewer is confident but not absolutely certain that the evaluation is correct

**Robotics Focus:**

Sufficient demonstration on hardware

**Strengths And Weaknesses:**

Strengths:
- The problem of grasp transfer across different robot grippers is important, novel, interesting, and challenging.
- The core idea of learning SDF-based 3D implicit representations for the object and gripper geometry jointly is quite new and interesting.
- The results of grasp transfers from human hand demonstrations to robot hand execution are interesting and good.

Weaknesses:
- The method learns one individual embedding for each input shape, and learns different embeddings for different shapes. Although the authors discussed this limitation and mentioned this is a future work, I think this is a quite big limitation, since it prevents the method from being used for a novel shape other than the 7 YCB objects. One needs to get the full 3D model, run the simulated grasps data collection process, and train all networks again. Firstly, it's not already possible to get a full 3D model. Secondly, if you got it, there are plenty of ways to analytically compute good grasps given a gripper, right?
- While I think the joint SDF learning for the object and the gripper is interesting, I don't quite get the motivations of learning a latnet embedding of grasps. The embedding is supervised using a heuristic of grasp similarity using different grippers -- the contacts over the object. Then, why do we need to learn such latent embedding? Why not directly compare the explicit contact heatmaps? The problem exaggerates when I find that the paper only performs grasp retrieval. You can compare the explicit contact heatmaps to do the retrieval, right? Can the authors show that you can solve for the grasp poses instead of retrieving one given a new gripper?
- While I agree that it's hard to define the similarity between two grasps using different grippers and I think the idea of using contact maps is reasonable, I think that many transferred grasps are quite different, for example, the grasp poses can be quite different while the contact regions are similar. I want to see more justifications for only using the contact maps as the criterion, and not using gripper pose similarity. Say, the easiest thing one can do is to compute the gripper geometry similarity? or the gripper skeleton chamfer distance?
- I don't think the authors can claim the data contribution as a main contribution since the grasps are synthetically generated.
- In Table 2, why "near GT" is similar to "near Z"? Also, as mentioned in Line 252, "has similar scores". This exaggerates my concern if learning a latent embedding is really better than directly comparing the explicit contact map similarity?
- The experiments are conducted over only 7 shapes, which is a very small scale.


**Summary Of Recommendation:**

I like the SDF part but don't understand the benefits of learning the latent embedding. Also, the method trains different individual embeddings for different shapes, considerably limiting its use case dealing with novel shapes. I have further concerns regarding the experiment scale, the grasp retrieval/transfer, and the claimed dataset contribution.

-- Post rebuttal
I've read the author's rebuttal and the other reviews. While I still remain concerned about training one embedding per shape, doing experiments over only seven shapes, and only using contact maps as the definition for grasp similarities, I do not object to accepting this paper as the paper's idea is generally interesting and novel, which is worth getting exposed to the community.

---

> ### Author Response · Authors · 2022-08-23
> **Responses to Reviewer hW79 - Part 1**
>
> We thank the reviewer for the critical feedback and valuable suggestions. We address the specific questions in two parts.
>
> **Q1:** Limitation of learning individual model for each input shape and analytically compute good grasps given a gripper.
>
> We agree with the reviewer that training a single network for each object is a limitation of our current method. In order to deal with a novel object, we have to train a different embedding network for it. We do not need to re-train the embedding networks for existing objects. As mentioned in our paper, we plan to explore learning a single embedding network for multiple objects as a future work.
>
> Regarding the reviewer’s concern on analytically computing good grasps given a gripper, we would like to point that our main goal in this work is not to synthetize grasps. Our method takes planned grasps of an object from multiple grippers, and embeds these grasps into a latent space. This implicit representation of grasps enables us to solve the following downstream tasks as illustrated in our experiments: shape reconstruction, grasp transfer and 6D object pose estimation with grasp contact optimization.
>
>
> **Q2:** Motivation for learning the latent embeddings in comparison to directly comparing the explicit contact heatmaps.
>
> We would like to mention that in addition to grasp retrieval, we have also shown that the grasp embeddings can be used for 6D object pose estimation with contact optimization and grasp transfer from human demonstrations in the real world. The goal of the grasp retrieval experiment is to verify that our learned grasp embeddings can enable the similarity between grasps from different grippers.
>
> The embeddings of grasps have two important properties in our method. First, a grasp embedding can be used to decode to the SDF of the grasping pose for an object. This property enables us to optimize the contact between a grasp and the object during 6D object pose estimation using a partial point cloud of the object. Second, the distance metric in the embedding space reserves the similarity between grasps, where the similarity is defined based on grasp contact maps. This enables us to perform transferring from human grasps to robot grasps.
>
> We agree with the reviewer that if we know the contact maps of all the grasps, we can use the contact maps for retrieval. But there are cases when the contact map is not available. In our grasp transferring experiment, a human demonstrates a grasp of an object. In this case, we do not have the contact map of the human grasp. Our solution is to use the point clouds of the human hand and the object to infer a latent grasp embedding, then we retrieve the closest robot grasp in the embedding space. An alternative way is to estimate the contact map using the estimated hand pose and object pose. In this case, the contact estimation method needs to deal with errors in pose estimation. There is no clear solution to this problem. Inferring a grasp embedding in the latent space helps us to circumvent some aspects of the contact map estimation problem in grasp transfer.
>
>
> **Q3:** Can the authors show that you can solve for the grasp poses instead of retrieving one?
>
> Our method embeds planned grasps into a common latent space. However, it cannot synthesize novel grasps in the latent space well. A randomly sampled vector in the grasp embedding space may not generate a meaningful grasp. We have mentioned this as a limitation of our method in the paper. Therefore, we perform grasp retrieval in the embedding space.

---

> > ### Author Response · Authors · 2022-08-23
> > **Responses to Reviewer hW79 - Part 2**
> >
> > **Q4:** Justifications for only using the contact maps as the criterion, and not using gripper pose similarity.
> >
> > Our motivation of using contact map similarity to define grasp similarity is from task-oriented grasping. People tend to grasp similar regions of an object when using the object for certain tasks. We agree with the reviewer that we can also add the gripper pose similarity in our method to define the grasp similarity. We can compute a weighted sum of the contact similarity and the gripper pose similarity. In general, our method can use different similarity metrics of grasps in learning the latent embeddings.
> >
> > **Q5:** I don't think the authors can claim the data contribution as a main contribution since the grasps are synthetically generated.
> >
> > A few grasping datasets in the literature use synthetically generated grasps in simulation such as the Dex-Net dataset by Mahler et al. and the ACRONYM dataset by Eppner et al. We will release the code and data of our multi-hand grasping dataset to the community.
> >
> > **Q6:** In Table 2, why "near GT" is similar to "near Z"? And is learning a latent embedding really better than directly comparing the explicit contact map similarity?
> >
> > This grasp retrieval experiment is to verify that the learned grasp embeddings preserve the similarity metric of using contact maps. Therefore, we should see “near Z” similar to “near GT” and “far Z” similar to “far GT”. As mentioned before, our latent embedding can be useful in grasp transfer of novel grasps, where we do not have the contact maps available.
> >
> > **Q7:** The experiments are conducted over only 7 shapes, which is a very small scale.
> >
> > Although we only conducted experiments on 7 YCB objects, each object has 250 grasps from 5 different grippers. Our method encodes all these grasps of an object into a single network. We will explore learning a unified network for multiple objects and multiple grippers in our future work.

---

### Official Review · Reviewer_z2Qy · 2022-07-30

**Originality:** Very Good
**Technical Quality:** Very Good
**Clarity Of Presentation:** Excellent
**Impact:** 4

**Recommendation:**

Strong Accept: I recommend accepting the paper and will argue for my recommendation even if other reviewers hold a different opinion.

**Summary:**

The paper proposes to use signed distance field to represent grasping behavior. The goal is to learn a representation that's transferrable across morphologies. The transfer happens during inference time by optimizing a latent representation. Awesome results are demonstrated by the authors.

**Issues:**

See my comment on related works.

**Quality Of The Limitations Section:**

Additional details required

**Reviewer Expertise:**

4: The reviewer is confident but not absolutely certain that the evaluation is correct

**Robotics Focus:**

Sufficient demonstration on hardware

**Strengths And Weaknesses:**

Strength:
* The paper is very well written. It's very easy to follow the paper, and the flow of the work develops really well. Many relevant concepts are introduced in a clear manner, which will be very helpful for readers of all kinds of backgrounds.
* The visualization are presented in a clean and informative ways. They are very helpful and demonstrate ideas clearly. I wish I can see this level of presentation in every paper.
* The method makes a lot of sense. It captures the essence of "how something is being grasped" really well. Using an test time inference / optimization pipeline makes a lot of sense and follows naturally when using an implicit representation.

Weakness:
* I think it would be great if more details on practicality is provided. this involved a few aspects:
- how fast can the transfer be performed? Is it possible to get in in real time? I think it might be challenging.
- how does the pipeline fit into a policy training perspective? currently we are transferring grasp of a single action on a single object. Can such pipeline be extended to a grasping policy?
* I think a few lines of related works are missing, which is in my opinion is weakest part of the paper. When introducing "Neural Implicit Representations." related works, wouldn't it make sense to introduce efforts that are using INR in robotics? namely:
line of work from UT Austin:
- Synergies Between Affordance and Geometry: 6-DoF Grasp Detection via Implicit Representations, Zhenyu Jiang, Yifeng Zhu, Maxwell Svetlik, Kuan Fang, Yuke Zhu, RSS, 2021
- ACID: Action-Conditional Implicit Visual Dynamics for Deformable Object Manipulation Bokui Shen, Zhenyu Jiang, Christopher Choy, Leonidas J. Guibas, Silvio Savarese, Anima Anandkumar, Yuke Zhu, RSS, 2022

line of work from MIT:
- Learning Models as Functionals of Signed-Distance Fields for Manipulation Planning, Danny Driess, Jung-Su Ha, Marc Toussaint, Russ Tedrake, CoRL 2021
- 3D Neural Scene Representations for Visuomotor Control, Yunzhu Li*, Shuang Li*, Vincent Sitzmann, Pulkit Agrawal, and Antonio Torralba, CoRL 2021

line of work from Google:
- VIRDO: Visio-tactile Implicit Representations of Deformable Objects., Youngsun Wi, Pete Florence, Andy Zeng, Nima Fazeli. ICRA 2022.
- NeRF-Supervision: Learning Dense Object Descriptors from Neural Radiance Fields. Lin Yen-Chen, Pete Florence, Jonathan T. Barron, Tsung-Yi Lin, Alberto Rodriguez, Phillip Isola. In ICRA 2022.

I really believe a wider coverage of the related work can only benefit the paper.

**Summary Of Recommendation:**

Awesome paper! Definitely should be an acceptance in my opinion. However, if the authors can do a better job with related work coverage, that would make the paper even better.

---

> ### Author Response · Authors · 2022-08-23
> **Responses to Reviewer z2Qy**
>
> Thank you for the detailed suggestion regarding related works and we have added them into the revised paper. We address specific questions below.
>
> **Q1:** How fast can the transfer be performed? Is it possible to get in in real time?
>
> In our experiments, the human hand detection and pose estimation run at 8fps. In order to infer the latent code of the human grasp, we need to perform an iterative optimization using the decoder network. This inference cannot be real time now. We perform 800 iterations for this optimization as in DeepSDF, which took about 7 seconds to infer the latent code of the human grasp. Less number of iterations can speed up the inference time, but it may produce inaccurate latent code. Addressing the challenges for real-time applications is certainly an interesting direction to pursue.
>
>
> **Q2:** How does the pipeline fit into a policy training perspective, especially for a grasping policy?
>
> One idea is to learn goal-conditioned policies for grasping. A goal-conditioned policy can take the latent code of a grasp as the grasping goal and the state representation of the environment as input, and output an action for grasping. For example, the action can be the 6D pose of the gripper. In real-world grasp transfer, the grasping goal can come from a human demonstration as in our current experiments. Learning grasping policies for multiple robotic hands is one the future work we will consider.

---

### Meta-Review · Area_Chair_GqEn · 2022-08-10

**Recommendation:** Accept (Poster)
**Confidence:** 5

**Metareview:**

All the three reviewers acknowledge somehow the novelty/originality of the paper. During the rebuttal, the authors bring in answers and clarifications to the concerns of R2 and R3. While they are not totally convincing, R2 has turned his recommendation from weak reject to weak accept, while R1 and R3 keep their rating. As a result, I recommend accepting the submission.


**Best Paper Nomination:**

No

---

> ### Author Response · Authors · 2022-08-23
> **Revised Paper**
>
> We thank the reviewers and the area chair for the valuable comments and suggestions. We have updated our paper according to the comments of the reviewers. The revised paper is attached in this reply.  We reply each reviewer for specific questions separately.